# Associations between Dynamic Vitamin D Level and Thyroid Function during Pregnancy

**DOI:** 10.3390/nu14183780

**Published:** 2022-09-14

**Authors:** Hui Wang, Hai-Jun Wang, Mingyuan Jiao, Na Han, Jinhui Xu, Heling Bao, Zheng Liu, Yuelong Ji

**Affiliations:** 1Department of Maternal and Child Health, School of Public Health, Peking University, Beijing 100191, China; 2Maternal and Child Health Care Hospital of Tongzhou District, Beijing 100191, China

**Keywords:** Vitamin D, deficiency, thyroid-stimulating hormone, free thyroxine, free triiodothyronine

## Abstract

Optimal Vitamin D (VitD) status and thyroid function are essential for pregnant women. This study aimed to explore associations between dynamic VitD status and thyroid function parameters in each trimester and throughout the pregnancy period. Information on all 8828 eligible participants was extracted from the Peking University Retrospective Birth Cohort in Tongzhou. Dynamic VitD status was represented as a combination of deficiency/sufficiency in the first and second trimesters. Thyroid function was assessed in three trimesters. The associations between VitD and thyroid function were assessed by multiple linear regression and generalized estimating equation models in each trimester and throughout the pregnancy period, respectively. The results indicated that both free thyroxine (fT4; β = 0.004; 95%CI: 0.003, 0.006; *p* < 0.001) and free triiodothyronine (fT3; β = 0.009; 95%CI: 0.004, 0.015; *p* = 0.001) had positive associations with VitD status in the first trimester. A VitD status that was sufficient in the first trimester and deficient in the second trimester had a lower TSH (β = −0.370; 95%CI: −0.710, −0.031; *p* = 0.033) compared with the group with sufficient VitD for both first and second trimesters. In conclusion, the associations between VitD and thyroid parameters existed throughout the pregnancy. Maintaining an adequate concentration of VitD is critical to support optimal thyroid function during pregnancy.

## 1. Introduction

Thyroid hormones are crucial for the maintenance of many fundamental functions in both adults and children [1]. During pregnancy, adequate thyroid hormone levels are essential for normal pregnancy and optimal fetal growth and development [2]. Maternal thyroid hormone is needed to support the development of placental and fetal hormones over the first half of pregnancy [3]. Since the fetal thyroid gland matures only after 18–20 weeks of gestation, before that, all thyroid hormones depend on maternal thyroxin (T4) transferred through the placenta [2]. Additionally, serum levels of thyroid-binding globulin (TBG) increase during pregnancy; this is consequently combined with free T4 (fT4) and free triiodothyronine (fT3) and then results in a slight decrease (10–15%) of these two hormones when pregnant women live in an area with sufficient iodine [4]. Meanwhile, increased urinary iodide clearance, thyroid hormone degradation, and human chorionic gonadotropin (hCG), which is a weak agonist of the thyroid-stimulating hormone receptor (TSH), all trigger higher thyroid hormone demand [5]. The total concentration of thyroxine should increase by around 20–50% to reach a euthyroid level during pregnancy [6]. Overall, pregnancy has a critical impact on the thyroid gland and its function.

Vitamin D (VitD) plays a pleiotropic role in the physiological and biological processes of the human body, such as cell growth, differentiation, maturation, anticarcinogenic effects, and anti-autoimmune activities, due to the wide expression of the VitD receptor (VDR) [7,8]. Recent studies have indicated that VitD deficiency is associated with thyroid disease since VDR is expressed in thyrocytes [9]. Vitamin deficiency has been associated with autoimmune thyroid disease (AITD), such as Grave’s disease (GD) and Hashimoto’s thyroiditis (HT) [10,11]. Studies have indicated that VitD deficiency is prevalent in China [12,13] and the prevalence of VitD deficiency in pregnant women is around 70% [14]. Furthermore, VitD deficiency or thyroid disorder during pregnancy has a similarly adverse impact on the pregnant woman and fetus, for example, preeclampsia, gestational diabetes, premature birth, and abnormal fetal mental development.

All the aforementioned phenomena indicate a potential association between VitD and thyroid function. Some studies have investigated the associations between VitD status and thyroid function during pregnancy; their general weakness was that they investigated the associations in the first or second trimester or in three trimesters with different participants [15,16,17]. Only one study explored the dynamic association between VitD status and thyroid function parameters across three trimesters among 50 pregnant women, which had relatively low statistical power [18]. Three out of four studies did not find associations between VitD and thyroid function, and only one study detected a positive association between VitD concentration and TSH in the second trimester. Therefore, the present study aimed to fill this research gap by using a relatively larger sample size to capture the dynamic associations between VitD status and thyroid function in the Peking University Retrospective Birth Cohort in Tongzhou.

## 2. Participants and Methods

### 2.1. Study Population

The present study is a retrospective cohort study. Information on all eligible participants was extracted from the Peking University Retrospective Birth Cohort in Tongzhou (39° N latitude) based on the hospital information system of Beijing [19]. The inclusion criteria were (1) singleton pregnancy; (2) maternal age between 18 and 49 years old; (3) the initial thyroid function test was performed in the first trimester; (4) without assisted reproduction; (5) without heart disease, hypertension, diabetes mellitus, kidney disease, and autoimmunity disease before pregnancy; (6) without family or personal history of thyroid disease; (7) information on the first pregnancy was retained if more than one pregnancy was detected for the same woman; and (8) verified last menstrual period between 28 October 2015 and 29 May 2019. Exclusion criteria were (1) levothyroxine (LT4) intake during pregnancy (*n* = 901); (2) no Vitamin D test for the first and second trimesters (*n* = 9290); and (3) absence of important covariables (*n* = 56). Finally, information on 8828 pregnant women who visited and delivered in the Tongzhou Maternal and Children Health Hospital and had vitamin tests twice was collected. Among them, 1552 pregnant women were tested for thyroid function in the first and second trimesters; 562 for thyroid function in the first and third trimesters; and 212 for the first, second, and third trimesters (Appendix A).

### 2.2. Data Collection

#### 2.2.1. Assessment of VitD Deficiency and Thyroid Function

Serum 25-hydroxyVitamin D [25(OH)D] was measured in the first (median 9.5 weeks of gestation) and second trimester (median 26.8 week of gestation) as the combination of 25(OH)D2 and 25(OH)D3 using the high-performance liquid chromatography mass spectrometry method. Thus, the combined value of 25(OH)D2 and 25(OH)D3 was used to represent the level of 25(OH)D. When 25(OH)D ≤ 20.0 ng/mL, then the corresponding participant was defined as having maternal VitD deficiency. We classified VitD status into four groups: deficient in the first trimester and sufficient in the second trimester (D1S2), deficient in the second trimester and sufficient in the first trimester (S1D2), sufficient in both trimesters (S1S2), and deficient in both trimesters (D1D2). This classification method could better reflect the dynamic VitD status in the first two trimesters. Furthermore, we categorized the first trimester VitD status into quartiles (quartile 1: median 10.3 nmol/L (full range 2.4–13.1); quartile 2: median 15.8 nmol/L (full range 3.1–18.3); quartile 3: median 21.3 nmol/L (full range 18.3–24.3); quartile 4: median 28.3 nmol/L (full range 24.3–92.9).

The thyroid function was assessed primarily during the first trimester of pregnancy. A small fraction of the pregnant women had three thyroid function tests during the pregnancy period. Fasting blood samples were taken from all participants between 8:00 a.m. and 10:00 a.m. Then, serum was used for testing TSH, free T4 (fT4), free T3 (fT3), and TPOAb via electrochemiluminescence immunoassays (ARCHITECT i2000, Abbott Core Laboratory, Abbott Park, IL, USA). According to the reference range provided by the manufacturer, TPOAb > 5.6 IU/mL was considered positive. All the reagents were matched with ARCHITECT i2000 and supplied by the same company. The intra and inter-assay coefficients of variation for the above-mentioned three indicators were all smaller than 10%.

#### 2.2.2. Assessment of Covariates

The following relevant sociodemographic and clinical variables were considered as covariates: maternal age, maternal educational level (high school or below, college, and university or above), employment status (yes, no), race (Han, other), parity history (primipara, multipara), folate supplementation status (yes, no), and VitD supplementation status (yes, no). Based on the data of the present study, 91% of pregnant women took VitD as a supplement. All demographic and medical information was collected by trained health professionals when the pregnant women initialized the perinatal files in the first trimester, including birth date, race, educational attainment, occupation, pregnancy history, disease history, and LMP. Body mass index (BMI) was calculated by using weight (kg) divided by the square of height (m^2^) with the first antenatal visit data and participants were classified as underweight (BMI < 18.5 kg/m^2^), normal weight (18.5 ≤ BMI < 24.0 kg/m^2^), obese (24.0 ≤ BMI < 28.0 kg/m^2^), or overweight (BMI > 28.0 kg/m^2^) according to the criteria issued by the Working Group on Obesity in China (WGOC) [20]. Since VitD is influenced by sunlight, we categorized the conception period (last menstrual period) into spring (February–May), summer (June–July), autumn (August–October) and winter (November–January) to generally represent the fluctuation of VitD concentrations.

### 2.3. Approval of Ethics

The study was approved by the Ethics Committee of the Peking University Health Science Center (IRB00001052-21023).

### 2.4. Statistical Analyses

Continuous variables were checked for normal distribution using the Kolmogorov–Smirnov test and were presented as mean (standard deviation, SD) for normally distributed data and as median (IQR) for skewed data. Continuous variables were compared by using one-way ANOVA test or Kruskal–Wallis rank sum test among the four subgroups. Categorical variables were presented as frequency (percentage), and compared using the Chi-square test or Fisher’s exact test among the four subgroups. Multiple linear regression was applied to assess associations of VitD status with thyroid function parameters at each trimester. Generalized estimation equation (GEE) analysis was applied to assess the associations of VitD status with thyroid function parameters throughout the pregnancy using identity link function and exchangeable correlation structure. The crude model was built without adjusting for any covariates, whereas the adjusted model was adjusted for all the covariates (maternal age, maternal educational levels, maternal employment status, parity, pre-pregnancy BMI class, Vitamin D supplement during pregnancy, gestational diabetes, and gestational hypertension disorder). The data analyses were performed using R software (version 4.1.1). A two-tailed *p*-value < 0.05 was considered statistically significant.

## 3. Results

### 3.1. Characteristics of the Study Population

The median value of maternal VitD (25(OH)D) was 18.3 ng/mL in the first trimester and 30.5 ng/mL in the second trimester (Table 1). The proportion of VitD deficiency reduced from 57.1% in the first trimester to 19.8% in the second trimester among total participants. In the present study, the prevalence of gestational diabetes mellitus was 28.6% and the gestational hypertensive disorder was 2.1%.

### 3.2. Characteristics of Thyroid Function with Different VitD Status

From the quartile analysis based on the first trimester, only fT3 increased steadily with higher VitD concentration in the first trimester (*p* < 0.001, Appendix A). Within the group of participants with two VitD measurements, the number of participants was the lowest in the S1D2 group (Table 2). In the first trimester, levels of fT3 and fT4 were different among those four groups (both *p* < 0.05). The highest median levels of fT3 and fT4 were detected in the S1D2 group. In the third trimester, the TSH levels were different among the four groups (*p* < 0.001), and the S1D2 group had the highest TSH level among these four groups.

### 3.3. Associations between VitD Status and Thyroid Function in Each Trimester Separately

In the first trimester, the contemporaneous VitD status was positively associated with fT3 and fT4 levels both in the continuous analysis (β = 0.004; 95%CI: 0.003, 0.006; *p* < 0.001 and β = 0.009; 95%CI: 0.004, 0.015; *P* = 0.001) and categorical analysis (β = −0.056; 95%CI: −0.097, −0.016; *p* = 0.007 and β = −0.196; 95%CI: −0.331, −0.060; *p* = 0.001), respectively (Table 3). In the second trimester, the contemporaneous VitD status was positively associated with fT3 (β = 0.003; 95%CI: 0.001, 0.006; *p* = 0.014). In the third trimester, the VitD status of the second trimester had a positive association with the fT4 level in the third trimester (β = 0.011; 95%CI: 0.001, 0.022; *p* = 0.030). However, the association between VitD status and TSH was more complex. The VitD level in the second trimester seems to have had a negative association with TSH level in the third trimester in the continuous analysis (β = −0.009; 95%CI: −0.016, −0.001; *p* = 0.024).

### 3.4. Associations between VitD Status and Thyroid Function across Three Trimesters

To further portray the associations between VitD and thyroid function throughout three trimesters, GEE analyses were conducted in 212 pregnant women who had thyroid parameters measured at each trimester (Table 4). S1D2 of VitD status had positive association with TSH (β = −0.370; 95%CI: −0.710, −0.031; *p* = 0.033), that is, compared with VitD that was sufficient in both the first and second trimester, the status of deficiency could reduce TSH. Since the sample size shrank substantially, therefore, further characteristics comparisons were performed between 212 pregnant women and the rest of the total population (Appendix A). It indicated that there is higher prevalence of TPOAb positivity (10.7% vs. 31.6%), GHD (2.0% vs. 4.2%), and GDM (28.0% vs. 37.7%) in the population who had three thyroid function tests (Appendix A).

## 4. Discussion

In the current study, we delineated the complex associations between VitD and thyroid function parameters in each trimester with the birth cohort data. The results indicated that fT4 and fT3 both had positive associations with VitD status in the first trimester. A VitD status that was sufficient in the first trimester and deficient in the second trimester had a lower TSH, compared with the group with sufficient VitD for both first and second trimesters. Previous research indicated that VitD status was negatively associated with GHD, GDM, and TPOAb positivity [21,22,23], and these diseases or syndromes were also associated with thyroid functions [24]. Therefore, we adjusted these factors rather than directly censoring them from the data.

The VitD levels during pregnancy showed large variation across nations and regions. Additionally, due to the different VitD supplementation recommendations within and between jurisdictions during pregnancy, the supplementation also varies among pregnant women. VitD deficiency is common in pregnant women, and the worldwide prevalence of VitD deficiency was 54% in pregnant women [25]. A study based on China Nutrition and Health Surveillance (CNHS) indicated that the prevalence of VitD deficiency (<20 ng/mL) was higher in 2015–2017 (87.43%) compared with 2010–2012 (73.4%) among pregnant women, regardless of VitD supplementation [26]. The prevalence of VitD deficiency was 30.57% in the Shanghai birth cohort in the first trimester, which was less than our current study (57.1%) [27]. A plausible reason might be that the altitude is higher in Beijing in comparison with Shanghai.

The natural process of pregnancy might also influence the levels of VitD. One study focusing on pregnant women who did not have VitD supplementation showed that the deficiency rate could reach 96% in the first trimester. Interestingly, the prevalence decreased to 78% and 76% in the second and third trimesters individually [18]. Therefore, it might indicate that there is a physiological increase in VitD during pregnancy, the lowest concentration of VitD was in the first trimester. In addition, this result was confirmed with a larger sample size in the Shanghai birth cohort in both the VitD-supplemented group and the non-supplemented group [27], and Indian pregnant women [28]. Although VitD concentration was substantially influenced by seasons, concentration trends were similar in different trimesters [28].

In the aspects of thyroid function parameters, the physiological changes of TSH, fT4, and fT3 are non-linear and intricate [2]. During the first 8–10 weeks of gestation, hCG concentration reaches the highest level and the α subunit of hCG is structurally similar to TSH, which increases the serum concentration of thyroxine, particularly in T4, and this in turn reduces the level of TSH *per se* through negative feedback via the hypothalamic–pituitary–thyroid axis [29]. Generally, for every 10,000 IU/L increasing in hCG, TSH decreases by 0.1 mU/L; TSH value drops to the valley at the gestation age of 10–12 weeks [30]. Therefore, during pregnancy, women normally have lower serum TSH concentrations than before pregnancy. After a first trimester, with the reduced concentration of hCG, the TSH concentration slowly increases and maintains at the plateau around 25 weeks of gestation. However, there are 10% and 5% fractions of women with a suppressed TSH in the second and the third trimester, respectively [29]. In contrast, the level of fT4 peaks around gestation age of 10–12 weeks and shifts downwards after this until birth [2]. To date, there have been only a few studies measuring fT3 through three trimesters with the gold standard measurement LC-MS/MS. According to the published data, the levels of fT3 synchronize with fT4 in late pregnancy [31,32]. Overall, the TSH, fT3, and fT4 levels should be dissected based on the different trimesters. Therefore, we drew a theoretical schematic diagram for VitD and thyroid parameter (Appendix A) for a better understanding.

In the present study, positive associations were detected between VitD concentration and fT3 and fT4 levels in the first trimester, individually. These results were logical and corresponded to physiological changes among pregnant women based on the aforementioned findings (Appendix A). In contrast, no association between VitD and thyroid functional parameters was detected in Zhao’s study with 50 Chinese women [18]. The plausible differences could be attributed two major aspects. First, the number of participants was only 50 in Zhao’s study, which generated relatively low R^2^ values in the multiple linear regression. Second, all participants did not supplement with VitD before or during pregnancy, which resulted in a generally low serum VitD level. Musa’s study was conducted with 132 Sudanese women in the first trimester and also did not find relationship between VitD level and thyroid parameters [16]. Both Zhao’s and Musa’s studies included participants who did not use VitD supplementation, although the non-supplementation period was different. The period of no supplementation of VitD in Musa’s study only accounted for the last six months before recruitment [16], while Zhao’s study recruited participants who did not use VitD or calcium supplementation before or during pregnancy. Nonetheless, the prevalence of VitD deficiency was significantly high in both Zhao’s (98%) and Musa’s (99.2%) cohorts.

Ahi et al. conducted correlation analyses between maternal VitD and thyroid function analysis with 66 Iranian pregnant women without VitD supplementation. The participants were approximately divided into three trimesters, and each trimester included 22 pregnant women. However, no statistical association was detected and the prevalence of VitD deficiency was 45.46% [15]. All of these results indicated that associations can only be detected with a certain amount of VitD. Intriguingly, Nizar carried out a study with sufficient levels of VitD (>30 ng/mL) in Ammaman-Jordan women and detected a negative association between VitD and TSH (r = −0.51, *p* < 0.005) [33]. However, Nizar did not indicate which trimester those participants were in when they conducted the study. We hypothesize that this scenario fits into the theoretical relationship between VitD and TSH in the first trimester (Appendix A). In the present study, we detect a positive association between VitD and TSH in the third-trimester model and the GEE model. These results indicated that the TSH level started to increase in the second trimester and mainly bounced up in the third trimester. Then, the overall average effect of VitD on TSH was positive during the entire pregnancy period. In addition, Pan et al. performed an association analysis with 277 pregnant women (without any thyroid-antibody positivity) in the second trimester; they revealed a positive association between TSH and VitD and negative associations between fT3/fT4 and VitD, respectively. In our present study, we revealed a positive association between fT3/fT4 and VitD in the first trimester. Overall, all those results fitted into the theoretical model in the first trimester (our present study) and second trimester (Pan’s study) individually (Appendix A).

Until now, the regulations between VitD and thyroid parameters have not been fully illustrated. The majority of studies have examined VitD deficiency and autoimmune thyroid disease (AITD) [9]. Since 1,25(OH)2D can suppress the adaptive immune system, it improves immune tolerance and represents a beneficial effect on a number of autoimmune diseases [34]. Additionally, animal studies have indicated that 1,25(OH)2D combined with cyclosporine could result in a synergistic effect to prevent the induction of experimental autoimmune thyroiditis (EAT) in CBA mice [35,36]. Furthermore, BALB/C mice could develop persistent hyperthyroidism after immunization with the TSH receptor only in VitD-deficient mice [37]. In rat experiments, researchers found that 1,25(OH)D3 could bind to the thyroid hormone receptors at the pituitary levels and modulated the secretion of TSH [38]. Kano et al. demonstrated that VitD concentration could be elevated after increasing T3, T4, and TSH in rats [39]. More related experiments should be conducted to further elucidate the mechanism of VitD and thyroid hormone interactions. Nonetheless, both adequate VitD level and euthyroidism are vital for fetal development. Therefore, VitD supplementation seems more important in the first trimester.

Several merits of the current study should be mentioned. It was the first study to reveal the associations between VitD and thyroid parameters across three trimesters with a relatively moderate population size, with the birth cohort data. Furthermore, we also dynamically considered Vitamin D concentration during the first two trimesters in terms of the high supplementation rate, which strengthened our findings. However, the present study also encountered limitations. Most pregnant women were only measured for their thyroid function in the first trimester routinely. Those participants who had more than one measurement of their thyroid functions had a high prevalence of gestational complications, such as GDM and GHD, which may mean the associations between VitD and thyroid function parameters were underestimated. Second, most participants did not have a VitD measurement in the third trimester, which made it difficult to portray the full view of VitD and thyroid function in our own dataset. Finally, it was an observational study in which causal inference ability was also limited.

In conclusion, the associations between VitD and thyroid parameters are dynamic. fT3 and fT4 were positively associated with VitD in the first trimester. TSH was positively associated with VitD, particularly in the third trimester. It is important to maintain an adequate level of VitD to support normal thyroid function from a nutritional point of view.

## Figures and Tables

**Table 1 nutrients-14-03780-t001:** Characteristics of participants in three different trimesters.

Characteristic	First Trimester, *n* = 8828	Second Trimester ^1^, *n* = 1396	Third Trimester ^2^, *n* = 562	*p* ^4^
Maternal age	29.40 (5.10) ^3^	29.20 (4.82)	29.75 (5.38)	0.033
Maternal education				0.894
Low (high school or below)	1308 (14.8)	197 (14.1)	77 (13.7)	
Middle (college)	3828 (43.4)	607 (43.5)	251 (44.7)	
High (university or above)	3692 (41.8)	592 (42.4)	234 (41.6)	
Maternal employment				0.194
Unemployed	980 (11.1)	140 (10.0)	72 (12.8)	
Employed	7848 (88.9)	1256 (90.0)	490 (87.2)	
Race/ethnicity				0.960
Han	8305 (94.1)	1316 (94.3)	529 (94.1)	
Other	523 (5.9)	80 (5.7)	33 (5.9)	
Parity				0.143
Primiparous	5049 (57.2)	785 (56.2)	343 (61.0)	
Multiparous	3779 (42.8)	611 (43.8)	219 (39.0)	
Maternal BMI class				0.130
Underweight	840 (9.5)	151 (10.8)	43 (7.7)	
Normal	5823 (66.0)	940 (67.3)	380 (67.6)	
Overweight	1716 (19.4)	247 (17.7)	106 (18.9)	
Obese	449 (5.1)	58 (4.2)	33 (5.9)	
Folate supplement				0.462
No	760 (8.6)	107 (7.7)	45 (8.0)	
Yes	8068 (91.4)	1289 (92.3)	517 (92.0)	
Multivitamin supplement				0.765
No	4290 (48.6)	671 (48.1)	265 (47.2)	
Yes	4538 (51.4)	725 (51.9)	297 (52.8)	
Season of conception				0.011
Spring	1974 (22.4)	301 (21.6)	134 (23.8)	
Summer	1631 (18.5)	257 (18.4)	133 (23.7)	
Autumn	2614 (29.6)	436 (31.2)	163 (29.0)	
Winter	2609 (29.6)	402 (28.8)	132 (23.5)	
Vitamin D (0–13 weeks), ng/mL	18.30 (11.20)	18.20 (11.70)	19.20 (10.67)	0.312
Vitamin D deficiency (0–13 weeks)				0.410
No	3783 (42.9)	600 (43.0)	257 (45.7)	
Yes	5045 (57.1)	796 (57.0)	305 (54.3)	
Vitamin D (14–28 weeks), ng/mL	30.50 (16.20)	30.55 (16.32)	30.50 (14.85)	0.905
Vitamin D deficiency (14–28 weeks)				0.124
No	7081 (80.2)	1106 (79.2)	468 (83.3)	
Yes	1747 (19.8)	290 (20.8)	94 (16.7)	
Vitamin D supplement				0.264
No	810 (9.2)	113 (8.1)	44 (7.8)	
Yes	8018 (90.8)	1283 (91.9)	518 (92.2)	
TPOAb positive (0–13 weeks)	993 (11.2)	234 (16.8)	105 (18.7)	<0.001
Gestational hypertensive disorder	185 (2.1)	26 (1.9)	27 (4.8)	<0.001
Gestational diabetes mellitus	2492 (28.2)	382 (27.4)	227 (40.4)	<0.001

^1^ Second-trimester participants should have both first- and second-trimester thyroid function tests irrespective of the third trimester thyroid function test. ^2^ Third-trimester participants should have both first- and third-trimester thyroid function tests irrespective of the second trimester thyroid function test. ^3^ Values are *n* (%) for categorical variables and median (IQR) for a continuous variable with a skewed distribution. Vitamin D concentration was measured with 25-hydroxyVitamin D in the serum. ^4^ Continuous variables were compared by using one-way ANOVA test for normal distribution data or Kruskal–Wallis rank sum test for skew distribution data among the four subgroups.

**Table 2 nutrients-14-03780-t002:** Parameters of thyroid function with different VitD status in three trimesters.

Characteristic	S1S2 ^1^	D1S2	S1D2	D1D2	*p* ^3^
First trimester (*n* = 8828)	*n* = 3324	*n* = 3757	*n* = 459	*n* = 1288	
TPOAb (IU/mL)	0.37 (0.66) ^2^	0.33 (0.59)	0.37 (0.67)	0.33 (0.53)	0.983
TSH (μIU/mL)	0.92 (0.91)	0.90 (0.97)	0.93 (1.01)	0.88 (0.97)	0.513
fT3 (pmol/L)	4.26 (0.62)	4.19 (0.61)	4.27 (0.64)	4.16 (0.64)	0.002
fT4 (pmol/L)	13.35 (2.05)	13.20 (2.08)	13.51 (2.03)	13.25 (2.04)	0.024
Second trimester (*n* = 1396)	*n* = 515	*n* = 591	*n* = 85	*n* = 205	
TSH (μIU/mL)	0.85 (0.82)	0.90 (0.81)	0.74 (0.90)	0.86 (0.87)	0.379
fT3 (pmol/L)	4.06 (0.60)	4.10 (0.60)	3.97 (0.49)	4.05 (0.59)	0.052
fT4 (pmol/L)	11.21 (1.70)	11.25 (1.76)	11.43 (1.91)	11.05 (1.86)	0.928
Third trimester (*n* = 562)	*n* = 225	*n* = 243	*n* = 32	*n* = 62	
TSH (μIU/mL)	1.27 (1.03)	1.31 (1.08)	1.69 (1.53)	1.25 (1.20)	<0.001
fT3 (pmol/L)	3.80 (0.66)	3.84 (0.66)	3.78 (0.52)	3.89 (0.60)	0.597
fT4 (pmol/L)	9.91 (1.78)	9.96 (1.74)	9.66 (1.46)	9.91 (1.85)	0.619

^1^ S1S2 means that VitD was sufficient in both the first and second trimester; D1S2 means that VitD was deficient in the first trimester and sufficient in the second trimester; S1D2 means that VitD was sufficient in the first trimester and deficient in the second trimester; D1D2 means that VitD was deficient in both the first and second trimesters. ^2^ Values are median (IQR) for a continuous variable with a skewed distribution. Vitamin D concentration was measured with 25-hydroxyVitamin D in the serum. ^3^ Continuous variables were compared by using one-way ANOVA test for normal distribution data or Kruskal–Wallis rank sum test for skew distribution data among the four subgroups.

**Table 3 nutrients-14-03780-t003:** Associations between VitD status and thyroid function parameter in each trimester ^1^.

Indicators	Vitamin D Status	First Trimester (*n* = 8828)			Second Trimester (*n* = 1396)			Third Trimester (*n* = 562)		
		β	95%CI	*p*	β	95%CI	*p*	β	95%CI	*p*
TSH	VitD level in 1st trimester	0.001	−0.001, 0.003	0.311	−0.001	−0.006, 0.003	0.547	−0.001	−0.011, 0.010	0.884
	VitD level in 2nd trimester	-	-	-	0.002	−0.001, 0.006	0.188	−0.009	−0.016, −0.001	0.024
	Vit D deficiency classification									
	D1S2	−0.016	−0.049, 0.018	0.365	−0.029	−0.116, 0.058	0.511	0.178	−0.011, 0.367	0.066
	S1D2	0.001	−0.040, 0.041	0.980	0.019	−0.081, 0.120	0.707	−0.378	−0.587, −0.169	<0.001
	D1D2	0.001	−0.045, 0.048	0.953	0.075	−0.037, 0.188	0.190	−0.455	−0.690, −0.221	<0.001
fT3	VitD level in 1st trimester	0.004	0.003, 0.006	<0.001	−0.001	−0.005, 0.002	0.512	−0.004	−0.010, 0.003	0.262
	VitD level in 2nd trimester	-	-	-	0.003	0.001, 0.006	0.014	−0.0001	−0.005, 0.005	0.962
	Vit D deficiency classification									
	D1S2	−0.027	−0.056, 0.003	0.076	−0.074	−0.138, −0.009	0.025	0.018	−0.099, 0.136	0.762
	S1D2	−0.014	−0.049, 0.021	0.436	0.001	−0.072, 0.075	0.972	0.055	−0.075, 0.184	0.410
	D1D2	−0.056	−0.097, −0.016	0.007	0.078	−0.004, 0.161	0.064	0.083	−0.063, 0.228	0.265
fT4	VitD level in 1st trimester	0.009	0.004, 0.015	0.001	0.009	−0.002, 0.020	0.106	0.001	−0.013, 0.016	0.842
	VitD level in 2nd trimester	-	-	-	0.001	−0.007, 0.009	0.849	0.011	0.001, 0.022	0.030
	Vit D deficiency classification									
	D1S2	0.077	−0.023, 0.176	0.130	−0.046	−0.241, 0.148	0.640	−0.090	−0.350, 0.170	0.498
	S1D2	−0.050	−0.168, 0.069	0.411	0.049	−0.175, 0.273	0.669	0.113	−0.175, 0.400	0.442
	D1D2	−0.196	−0.331, −0.060	0.005	0.013	−0.238, 0.264	0.921	0.146	−0.176, 0.469	0.374

^1^ Multiple linear regression was applied to assess the associations of the Vitamin D status with thyroid function parameters at each trimester. Adjusted for maternal age, maternal educational levels, maternal employment status, parity, prenatal BMI class, Vitamin D supplement, folate and multivitamin supplementation, GHD, GDM, TPOAb positivity during pregnancy, and seasons of conception.

**Table 4 nutrients-14-03780-t004:** Associations between VitD status and thyroid function parameter across three trimesters ^1^ (*n* = 212).

		Unadjusted Analysis			Adjusted Analysis ^2^		
Indicator	VitD Status	Beta	95%CI	*p*	Beta	95%CI	*p*
TSH	VitD level in 1st trimester	0.001	−0.012, 0.014	0.855	0.004	−0.009, 0.018	0.519
	VitD level in 2nd trimester	0.002	−0.008, 0.012	0.696	0.001	−0.008, 0.011	0.786
	VitD deficiency classification						
	D1S2	−0.155	−0.367, 0.057	0.152	−0.133	−0.358, 0.091	0.245
	S1D2	−0.283	−0.586, 0.020	0.067	−0.370	−0.710, −0.031	0.033
	D1D2	−0.050	−0.422, 0.323	0.794	−0.215	−0.632, 0.202	0.312
FT3	VitD level in 1st trimester	−0.012	−0.025, 0.002	0.089	−0.015	−0.030, 0.000	0.053
	VitD level in 2nd trimester	−0.002	−0.011, 0.008	0.716	−0.002	−0.014, 0.009	0.695
	VitD deficiency classification						
	D1S2	0.010	−0.232, 0.252	0.935	0.072	−0.201, 0.344	0.605
	S1D2	−0.010	−0.238, 0.218	0.933	−0.054	−0.309, 0.201	0.678
	D1D2	0.193	−0.020, 0.407	0.076	0.181	−0.084, 0.445	0.181
FT4	VitD level in 1st trimester	−0.021	−0.054, 0.013	0.222	−0.030	−0.067, 0.008	0.121
	VitD level in 2nd trimester	0.000	−0.0278, 0.0287	0.975	−0.004	−0.034, 0.027	0.819
	VitD deficiency classification						
	D1S2	0.546	−0.413, 1.504	0.265	0.755	−0.167, 1.678	0.109
	S1D2	0.781	−0.066, 1.627	0.071	0.853	−0.009, 1.715	0.052
	D1D2	0.760	0.043, 1.477	0.038	0.799	−0.023, 1.622	0.057

^1^ Generalized estimation equation (GEE) analysis was applied to assess associations of the Vitamin D status with thyroid function parameters throughout pregnancy using identity link function and exchangeable correlation structure. ^2^ Adjusted for maternal age, maternal educational levels, maternal employment status, parity, pre-pregnancy BMI class, Vitamin D supplement, folate and multivitamin supplementation, positivity for GHD, GDM, TPOAb during pregnancy, and seasons of conception.

## Data Availability

The data presented in this study are available on request from the corresponding author.

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
