# Peer review of "Associations between Dynamic Vitamin D Level and Thyroid Function during Pregnancy"

_nutrients, 2022, doi:10.3390/nu14183780_

Round 1

Reviewer 1 Report

139 / 5.000 Importante work for the area, methodology presented correctly, well-defined working hypothesis and apropriate conclusions. Some grammar and spelling errors must be corrected. 

Resultados de tradu

Author Response

Response: Thanks for your valuable comment. We did a thorough check for grammar and spelling errors.

Reviewer 2 Report

Well.written and designed article the authors have managed to support the role of vit d and its relation with thyroid function during pregnancy in a large cohort of patients

Author Response

Response: Thanks for the comment.

Reviewer 3 Report

Considering that an optimal vitamin D status and  thyroid function are essential in pregnant women,, Wang and co-workers investigated retrospectively the association between these parameters by appropriate statistical methods in three trimester of pregnancy in a large population of pregnant women. They found a positive correlation between free thyroid hormones and vitamin D in the first trimester, whereas a   vitamin D status sufficient in the first trimester but deficient in the second trimester was associated with a lower TSH. They conclude that mantaining an adequate concentration of Vitamin D   is fundamental  to ensure an optimal thyroid function throughout the whole pregnancy.

Even if the results of this paper do not add further significant news on the role and the importance of vitamin D and thyroid function,I think they are of some interest suggesting a careful surveillance of the association between these parameters during pregnancy to allow an  early therapeutic correction in case of occurrence of some deficiency. I would suuggest reducing the number of tables  and streamlining the Introduction and Discussion paragraphs to improve the readibility of the paper

- Some other recent review and meta-analysis on this subject should be cited. For example : Taheriniya et al. Vitamin D and thyroid disorders: a systematic review and meta-analysis of observational studies. BMC Endocr Disord, 2021

Author Response

 I would suuggest reducing the number of tables  and streamlining the Introduction and Discussion paragraphs to improve the readibility of the paper.

Response: your suggestions are well received. We improved the introduction and discussion parts accordingly. Page 1-2 and pages 8-9. In terms of tables, we would like to remain them as it is better for us to illustrate the associations step by step.

- Some other recent review and meta-analysis on this subject should be cited. For example : Taheriniya et al. Vitamin D and thyroid disorders: a systematic review and meta-analysis of observational studies. BMC Endocr Disord, 2021

Response: Thanks for your suggestion, it is a valuable document to read and learn. Wee cited this paper on page 2, which is numbered as reference 10.